# Clear Cell Acanthoma with Malignant Cytologic Features: A Case Report and Review of the Literature

Gabriella Melson [1], Elie Saliba [1,2], Shreya Patel [3], Richard Eisen [3] and Candice E. Brem [1,*]

1   Section of Dermatopathology, Department of Dermatology, Boston University Chobanian & Avedisian School of Medicine, Boston, MA 02118, USA
2   Department of Dermatology, Brown University, Providence, RI 02903, USA
3   Department of Dermatology, Boston University Chobanian & Avedisian School of Medicine, Boston, MA 02118, USA
*   Correspondence: candice.brem@bmc.org

**Abstract:** Clear cell acanthoma (CCA) is classically considered a benign epidermal tumor, although rare case reports have described CCA with malignant features. Here, we present a case of a patient with a biopsy proven CCA that regrew post-biopsy and was subsequently completely excised. Histologic examination of the tumor in the excision specimen revealed malignant cytologic features that were not present in the initial biopsy. A review of the literature identified five additional cases of CCA with similar malignant cytologic features. On analysis, common histopathologic characteristics included cellular pleomorphism, increased nuclear-to-cytoplasmic ratio, prominent nucleoli, and atypical mitotic figures. We support the designation of atypical clear cell acanthoma for these entities with features of both CCA and significant cytologic atypia. As none of these cases exhibited clinically aggressive behavior, further study is warranted.

**Keywords:** clear cell acanthoma; clear cell acanthocarcinoma; malignant clear cell acanthoma; CCA





## 1. Introduction

Clear cell acanthoma (CCA) is an epidermal tumor composed of pale-staining, glycogenated keratinocytes with a sharp demarcation between the tumor and surrounding uninvolved epidermis. Although CCA is considered a benign tumor, rare case reports have described CCA with malignant features under various names, including atypical CCA, malignant CCA, and squamous cell carcinoma in situ (SCCIS) arising within CCA. Here, we present a case of a patient with a biopsy proven CCA that regrew and was subsequently excised. Histopathologic examination of the tumor in the excision specimen revealed malignant cytologic features that were not present in the initial biopsy.

## 2. Case Report

A 78-year-old male with prior history of immune thrombocytopenia and multiple nonmelanoma skin cancers presented to dermatology clinic with a new growth on the thenar eminence of the left palm (Figure 1A) measuring 1.0 cm × 0.8 cm. He reported it had been present for several weeks and was occasionally painful. It persisted after treatment with liquid nitrogen. At his next visit, a tangential biopsy was performed. Histopathology showed marked epidermal hyperplasia with areas of well-demarcated pallor and mild spongiosis consistent with CCA (Figure 1B,C).

Four months after the initial biopsy, the patient returned to dermatology clinic with regrowth of the tumor at the same location. On examination there was a 0.7 cm by 0.8 cm erythematous, friable papule on the thenar eminence of the left palm (Figure 2A). Dermoscopy revealed glomerular vessels in a reticular pattern and few punctiform vessels, characteristic of CCA (Figure 2B). During the two-month period following this regrowth, the lesion was not treated, and it remained unchanged. Six months after the initial biopsy,

and approximately seven months post treatment with liquid nitrogen, the papule was excised with a 0.2 cm margin.

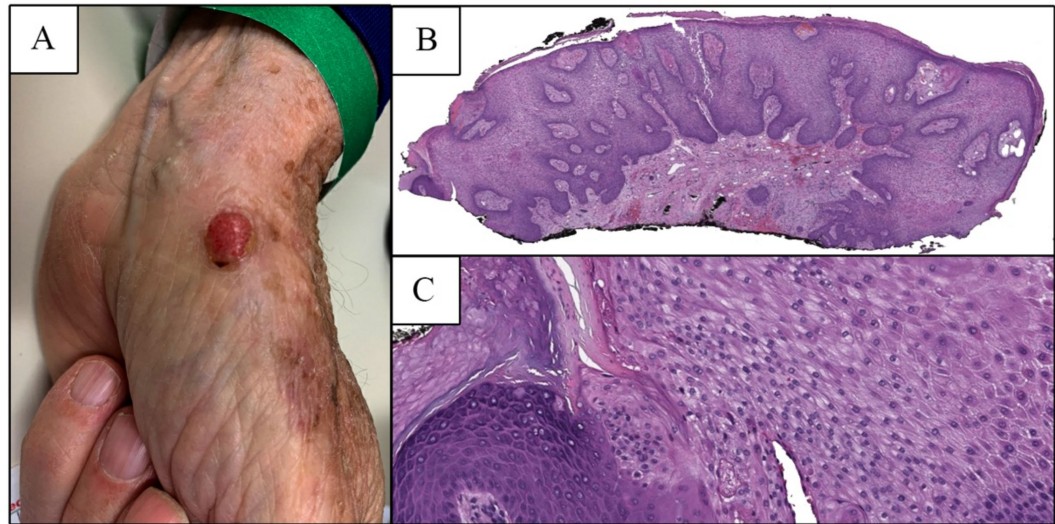

**Figure 1.** (**A**) Clinical image of the initial left thenar eminence lesion, prior to shave biopsy. (**B**) Low power examination of the initial shave biopsy specimen revealed an epidermal tumor exhibiting psoriasiform hyperplasia composed of pale staining cells (H&E, 20×) with (**C**) an abrupt transition to the adjacent normal epidermis at one lateral edge (H&E, 100×).

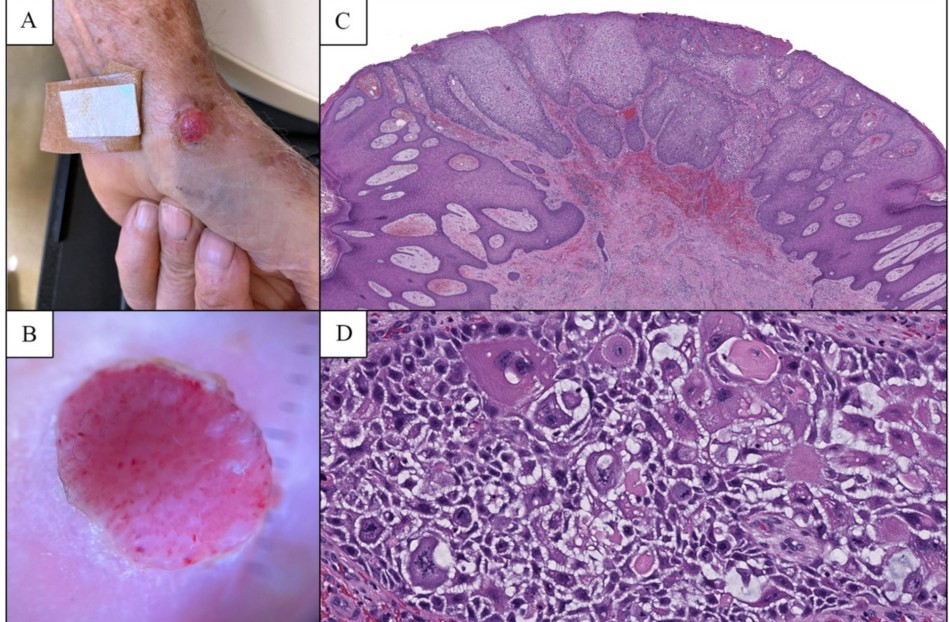

**Figure 2.** (**A**) Clinical image of the regrown lesion on the left thenar eminence after initial biopsy, appearing similar to the original lesion in Figure 1A. (**B**) Dermoscopic examination of this lesion showed glomerular vessels in a reticular pattern. (**C**) Low power examination of the subsequent excision specimen revealed an epidermal tumor composed of pale-staining cells with sharp lateral demarcation from the surrounding skin (H&E, 20×). (**D**) Pleomorphism with mono-and multinucleated keratinocytes and numerous mitotic figures, including atypical forms, were identified (H&E, 200×).

Histologic examination of the excision specimen demonstrated irregular epidermal hyperplasia composed of glassy to pale-staining keratinocytes (Figure 2C) exhibiting significant nuclear pleomorphism. Higher power examination toward the central/deep aspect of

the lesion revealed mono-and multinucleated keratinocytes as well as numerous mitotic figures (10 per 1 high-powered field), many of which were atypical (Figure 2D). Malignancy was favored given the degree of nuclear pleomorphism and the presence of atypical mitotic figures. Based on the histopathologic findings, as well as the patient's clinical history and prior biopsy result, a diagnosis of CCA with malignant cytologic features was made. The lesion has not recurred nine months post-excision.

## 3. Discussion

CCA was originally described by Degos et al. in 1962 as a benign epidermal tumor composed of pale, glycogenated keratinocytes with an abrupt cut-off between the tumor and uninvolved epidermis [1]. Additional reported findings include psoriasiform epidermal hyperplasia, sparing of adnexal epithelium, mild spongiosis, and exocytosis of neutrophils [2]. Despite extensive investigation of the histopathologic features of CCA, histogenesis and etiology remain unclear. Indeed, the cell of origin has been the subject of debate with various theories proposing epidermal, eccrine, or pilar origin [3,4]. Additionally, although CCA was initially described as a true neoplasm, there is some evidence to suggest that CCA may be inflammatory in etiology and could represent a psoriasiform reaction pattern [5,6]. Since its initial description, various subtypes of CCA have been reported, including giant [7], polypoid [8], and pigmented variants [9].

CCAs with malignant features have only rarely been reported in the literature (Table 1). Grunwald et al. reported the first two cases of clear cell acanthoma with atypical features in 1991 [10]. Both cases exhibited nuclear pleomorphism, a high nuclear-to-cytoplasmic ratio, prominent nucleoli, and atypical mitotic figures in addition to classic features of CCA. Neither local recurrence nor metastasis occurred in either case with a follow-up period of four to five years. Lin et al. subsequently reported a case of CCA with malignant features [11]. This lesion was composed of background anastomosing psoriasiform hyperplasia consistent with CCA; however, the lesional cells were cytologically atypical, demonstrating nuclear pleomorphism, prominent nucleoli, and mitotic figures. This tumor was believed to represent the malignant counterpart of CCA and was termed "malignant CCA." The tumor had not recurred six-months post- operatively. A case of CCA with architectural features of keratoacanthoma has also been reported [12]. While no marked cytologic atypia or atypical mitotic figures were noted in this case, the authors interpreted their findings as malignant degeneration, with features of keratoacanthoma, within a large CCA.

Additionally, there have been rare case reports of CCA associated with squamous cell carcinoma in situ (SCCIS). Parsons and Ratz reported a case of SCCIS arising within a clear cell acanthoma [13]. This case featured severe, full thickness cytologic atypia of epidermal keratinocytes with background changes characteristic of CCA. The authors note that this atypia could represent malignant degeneration of clear cell acanthoma rather than coincidental SCCIS. Multiple clear cell acanthomas were also reported in a patient with a history of multiple lesions of SCCIS in the setting of suspected arsenic exposure [14]. One CCA was reported to have arisen at the site of a previously excised SCCIS. While the authors raise the possibility of CCA representing an early stage of SCCIS or that arsenic exposure could predispose individuals to the development of both CCA and SCCIS, this could also support the theory that CCA may be an inflammatory reaction pattern.

**Table 1.** Summary of previously reported cases of CCA with malignant features.

| Case | Clinical Information | | | | | | Atypical Histopathologic Features | | | | | Interpretation | Follow-Up |
|---|---|---|---|---|---|---|---|---|---|---|---|---|---|
| | Patient Age/Sex | Location | Size * | Description | Duration | Initial Biopsy of CCA | Pleomorphism | Increased NC Ratio | Prominent Nucleoli | Atypical Mitotic Figures | | Interpretation | Follow-Up |
| Grunwald et al., 1991 [10] | 74/M | Forehead | 1 cm | Dome-shaped, scaly | 2 years | N/A | + | + | + | + | | Atypical CCA | No recurrence or metastases at 4–5 years [†] |
| | 71/F | Forehead | 1.5 cm | Slightly elevated, pink | 3 years | N/A | + | − | + | + | | Atypical CCA | |
| Parsons and Ratz, 1997 [13] | 60/F | Right leg | 2.6 cm | Fungating erythematous tumor | 2 years | + | + | NR | NR | NR | | SCCIS arising within CCA | Mohs surgery, NED at 6 months |
| Arida et al., 2006 [12] | 45/M | Left groin | 3 cm | Reddish polypoid lesion | Many years | N/A | − [‡] | − | − [§] | NR | | Malignant degeneration within CCA | Excision [†], follow-up NR |
| Lin et al., 2016 [11] | 92/F | Temple | 1.5 cm | Erythematous moist nodule | 3 months | N/A | + | + | − | NR [¶] | | Malignant CCA | Biopsy and cryotherapy, NED at 6 months |
| Current case | 78/M | Left palm | 1 cm | Erythematous friable papule | weeks | + | + | + | + | + | | Atypical CCA | Complete excision, NED at 9 months |

CCA, Clear cell acanthoma; N/A, not applicable; NED, no evidence of disease; NC ratio, nuclear-to-cytoplasmic ratio; NR, Not reported; SCCIS, squamous cell carcinoma in situ; +, present; −, not present. * Largest dimension of the lesion reported. [†] Completeness of biopsy/excision not reported. [‡] Case report mentions only slightly enlarged nuclei. [§] Case report mentions only small nucleoli. [¶] Case report mentions mitotic figures but does not specify atypical forms.

Overall, differentiating SCCIS arising within CCA, clear cell SCCIS and malignant degeneration of CCA remains a diagnostic challenge. The keratinocytes in CCA appear pale due to the presence of glycogen, which is highlighted by PAS stain and is diastase sensitive [15]. Previously, some authors have reported an absence of glycogen in clear cell SCCIS, raising the possibility that PAS may have diagnostic utility in differentiating clear cell SCCIS from CCA [16–18]. However, in contrast, a 2007 study of SCCIS with clear cell change showed that the keratinocytes in these cases contained numerous PAS-positive, diastase-sensitive granules as well as cytokeratin expression consistent with outer root sheath differentiation [19]. These results suggest that PAS staining pattern cannot reliably differentiate between SCCIS with clear cell change and malignant CCA.

In our case, as the malignant cytologic features were present only in the excision specimen of a previously biopsied CCA with regrowth, a component of a reactive epidermal process including pseudoepitheliomatous epidermal hyperplasia was considered. However, given the degree of cytologic atypia [20] and presence of atypical mitotic figures [21], a diagnosis CCA with malignant cytologic features was favored.

Nevertheless, as all previously reported cases of CCA with malignant cytologic features to date have exhibited clinically benign behavior, with no recurrences or metastases, this lesion is perhaps best considered an atypical CCA. While further study to determine the true biologic behavior of this entity is warranted, given the rarity of similar reported cases, it appears that excision may be curative.

**Author Contributions:** Conceptualization, G.M., E.S., S.P., R.E. and C.E.B.; writing—original draft preparation, G.M., E.S., S.P. and C.E.B.; writing—review and editing, R.E. and C.E.B. All authors have read and agreed to the published version of the manuscript.

**Funding:** This research received no external funding.

**Institutional Review Board Statement:** Not applicable.

**Informed Consent Statement:** Written informed consent has been obtained from the patient(s) to publish this paper.

**Data Availability Statement:** Not applicable.

**Conflicts of Interest:** The authors declare no conflict of interest.

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
