# Peer review of "Clear Cell Acanthoma with Malignant Cytologic Features: A Case Report and Review of the Literature"

_dermatopathology, doi:10.3390/dermatopathology9040041_

Round 1
Reviewer 1 Report
The authors present a well-written case report of the rare entity of clear cell acanthoma with malignant features. The pictures are of high quality and a comprehensive table of previously published case reports is added to allow better comprehension of the topic. In the discussion all relevant aspects of the topic including ambiguities and the somewhat misleading nomenclature is included. I favor publication, thanks for the excellent contribution of this interesting case.
Author Response
The authors wish to thank the reviewer for their insightful comments and review of our manuscript.

Reviewer 2 Report
This is a case report of a tumour diagnosed as clear cell acanthoma (CCA) with malignant (cytological) features.
The cases previously published as ‘malignant CCA’ or ‘atypical CCA’ cases, are in my view questionable, as on careful examination of their microscopic figures, most of them seem to correspond to clear-cell Bowen’s disease or squamous cell carcinoma in situ from clear cells. The case reported here shows malignant cytological features on the ‘recurrent’ lesion but has no unequivocal features of malignancy (no dermal, perineural or lymphovascular invasion, no metastasis or death); therefore this case may at best be described as atypical CCA (but not really as malignant CCA or acanthocarcinoma from clear cells). I therefore believe this case is a CCA with atypical cytological features, and encourage the authors to adopt this term atypical. In the same line of thought, the title should mention ‘malignant cytological features’ instead of ‘malignant features’ (which implies clinical malignancy).
Other issues that merit improvement/clarification:
- it seems that the initial lesion was not totally excised, therefore the lesion that was subsequently excised is not a recurrence but rather persisting, residual tumor. Therefore the term ‘recurrence’ is unsuitable for the totally-excised lesion.
- the malignant cytologic features were observed only in the 2nd excision of the tumor but not in the initial biopsy. I suspect that the initial tangential biopsy may have played a role in the development of ‘malignant’ cytologic features seen in the 2nd excision (which could correspond to reactive atypia). Please comment on this possibility. In this respect, it would have been interesting to study the 1st and 2nd excisions with immunohistochemistry, eg for Ki67 and p53, to better support the ‘malignant’ features of the proliferating keratinocytes. PAS staining could also be performed to assess a potential difference between the atypical and typical CCA cells.
- it is mentioned that the lesion ‘persisted after treatment with liquid nitrogen’. How long before had this treatment been applied? Can the authors rule out the role of prior cryotherapy to the development of atypical cytological features?
- the size of the initial tumour could be mentioned.
- “During the two-month period following recurrence, the lesion was not treated, and it remained unchanged”: the site of this sentence is confusing. This sentence should be inserted before the sentence describing total excision.
- Specifiy which area in panel 2C corresponds to panel 2D as this is not obvious in the low magnification provided.
Author Response
The authors wish to thank the editors and reviewer 2 for their review of the following manuscript: Dermatopathology-1875676. Please see the attached PDF document. Thank you

Round 2
Reviewer 2 Report
Thank you for having replied to most of my concerns. I find however the lack of immunohistochemistry and PAS staining (honestly quite simple techniques) regrettable, as the results could have rendered this case much more informative/interesting.
Author Response
The authors wish to thank reviewer 2 for their time and speedy response. Please see the attached PDF document.
